# Immunogenicity Analysis of PCV3 Recombinant Capsid Protein Virus-like Particles and Their Application in Antibodies Detection

**DOI:** 10.3390/ijms241210377

**Published:** 2023-06-20

**Authors:** Xuyang Cao, Min Huang, Ying Wang, Yanzhi Chen, Hanwen Yang, Fusheng Quan

**Affiliations:** 1College of Veterinary Medicine, Northwest A&F University, Yangling 712100, China; caoxuyang@nwsuaf.edu.cn (X.C.); huangmin123@nwsuaf.edu.cn (M.H.); wangying2023@nwafu.edu.cn (Y.W.); chenyanzhi@nwafu.edu.cn (Y.C.); 2020050603@nwafu.edu.cn (H.Y.); 2Key Laboratory of Animal Biotechnology, Ministry of Agriculture, Yangling 712100, China

**Keywords:** porcine circovirus type 3 (PCV3), capsid protein (Cap), virus-like particles (VLPs), enzyme-linked immunosorbent assay (ELISA)

## Abstract

Porcine circovirus type 3 is a newly emerging pathogen of porcine circovirus associated disease (PCVAD). Currently, there is no commercially available vaccine, resulting in huge economic losses to the pig industry. Porcine circovirus type 3 capsid protein (Cap) can self-assemble into virus-like particles (VLPs). Therefore, the expression of the recombinant Cap protein is of great significance for the prevention, diagnosis and control of porcine circovirus type 3 associated diseases. In this study, the recombinant Cap protein was successfully expressed in *Escherichia coli* by deleting the nuclear localization sequence (NLS). The VLPs were observed by transmission electron microscopy. To evaluate the immunogenicity of the recombinant Cap protein, mice were immunized. As a result, the recombinant Cap protein can induce higher levels of humoral and cellular immune responses. A VLP-based ELISA method was developed for the detection of antibodies. The established ELISA method has good sensitivity, specificity, repeatability and clinical applicability. These results demonstrate the successful expression of the PCV3 recombinant Cap protein and the preparation of recombinant Cap protein VLPs, which can be used for the preparation of subunit vaccines. Meanwhile, the established I-ELISA method lays a foundation for the development of the commercial PCV3 serological antibody detection kit.

## 1. Introduction

Porcine circovirus (PCV) is a small non-enveloped virus belonging to the Circovirus genus of the Circoviridae family. Its virion diameter is about 14 to 17 nm, making it one of the smallest animal viruses known so far [1]. PCV includes four genotypes, namely PCV1, PCV2, PCV3 and PCV4, which are very similar in structure and composition [2]. PCV1 was first discovered in PK-15 cells in 1974 and is not pathogenic [3]. PCV2, a major pathogen, was discovered in Canada in 1995 [4]. PCV4 was first discovered in Hunan Province, China in 2019, and its infection can cause severe clinical symptoms such as respiratory disease and diarrhea in pigs, but its pathogenic mechanism remains unclear [5]. PCV3 was also discovered in the United States in 2016, and this viral disease can cause porcine dermatitis and nephropathy syndrome (PDNS), sow abortion, piglet heart and multi-system inflammation, and has been widely found around the world [6]. Similar to other circoviruses, the full length of the PCV3 genome is 2000 bp and contains three inverted open reading frames, ORF1, ORF2 and ORF3. The full length of ORF1 encoding replicase protein (Rep) is 896 bp and encoding 296 aa. The full length of ORF3 is 693 bp, the code is 230 aa and the specific function is not clear. ORF2 encodes nucleocapsid protein (Cap), which has a full length of 645 bp and encodes 214 aa. It is the only structural protein of PCV3 and can self-assemble into virus-like particles (VLP) that are morphographically similar to PCV3 virions. The genome of PCV3 has been sequenced, but the homology with other porcine circoviruses is low, and the sequence homology with PCV2 is only 37% to 40%. There have been no reports of cross-protection against PCV3 with a commercially available PCV2 vaccine.

The VLP has no viral genetic material, cannot replicate and does not have the ability to infect, with a similar spatial three-dimensional structure of natural virus particles, which can effectively induce an immune protection effect in the organism [7]. At the same time, VLPs are more suitable for the development of serological diagnostic tests because of their higher safety and ease of use. An enzyme-linked immunosorbent assay (ELISA) based on VLPs is widely used to measure antibodies or neutralizing epitopes. For example, PCV2 VLPs are used as coated antigens so that serum-neutralizing antibodies can be detected in the ELISA [8]. Therefore, the PCV3 Cap protein has become an ideal target for the in vitro expression and development of subunit vaccines and related detection methods.

Pro-inflammatory cytokines play a key role in the development and maintenance of inflammation and can lead to multiple organ damage. Previous studies have shown that PCV3 infection up-regulates pro-inflammatory cytokines in pigs [9]. Moreover, the PCV3-mediated clinical symptoms of illness and tissue damage may be caused by the high levels of pro-inflammatory cytokines. Recent studies have also confirmed that PCV3 infection does cause an inflammatory response, in which IFN-γ, IL-6, IL-8, TNF-α and other cytokines are significantly increased after infection [10]. In this study, we successfully obtained the truncated expression of the recombinant Cap protein through the *E. coli* prokaryotic expression system, and successfully assembled it into VLPs in vitro. Antibody levels and interferon (IFN-γ), IL-4, IL-2 and T cell subsets were evaluated by immunizing mice to understand the potential application of the recombinant Cap protein as a candidate vaccine based on VLP. This study also developed a highly specific, sensitive and repeatable VLP-based I-ELISA method. This method has great significance for screening PCV3 antibodies in the clinic.

## 2. Results

### 2.1. Prokaryotic Expression of Recombinant Cap Protein

The recombinant pET32a-Cap protein was expressed, and it was found by SDS-PAGE and Western blotting. The recombinant protein was expressed in the precipitate, but not in the supernatant, mainly in the form of inclusion bodies with a size of about 25 KD (Figure 1). After purification by the nickel column, the purified recombinant Cap protein was found by SDS-PAGE (Figure 2B). To obtain the recombinant Cap protein with high expression, the IPTG-induced concentration screening showed that the protein expression gradually increased from 0.8 mM to 1.2 mM (Figure 2A). The obtained purified protein was concentrated, and the desired target protein was obtained by SDS-PAGE and a WB analysis (Figure 3).

### 2.2. Recombinant Cap Protein Can Be Assembled into VLPs In Vitro

After the purified recombinant Cap protein was dialyzed in dialysate, the TEM analysis showed that the recombinant Cap protein could be assembled into VLPs with a diameter of about 20 nm in vitro (Figure 4).

### 2.3. Recombinant Cap Protein Can Induce Specific Antibody Response in Mice

The plate was coated with the purified recombinant Cap protein. The ELISA analysis indicated that the mice immunized with a recombinant Cap protein showed an upward trend from the third week on. The antibody titer OD_450_ reached above 3.0, with no response in the control group. The results indicate that the recombinant protein had good immune effects (Figure 5).

### 2.4. Mice Immunized with Recombinant Cap Protein Produced Higher Antibody Levels

The ELISA results showed that after immunizing mice with the recombinant Cap protein, the antibody dilution could reach about 200,000 times and S/N could reach more than 60. It was indicated that mice immunized with a recombinant Cap protein produced a high level of antibody response (Figure 6).

### 2.5. Recombinant Cap Protein Can Induce T Cell Immune Response in Mice

The study further analyzed the cellular immune response induced by recombinant Cap proteins. The percentages of CD3^+^T cells and CD3^+^ CD4^+^ cells in mouse peripheral blood monocytes (PBMC) were significantly different in the experimental group compared with the control group (*p* < 0.05). A further analysis of CD3^+^ CD8^+^ T cell subtypes showed a higher significance in the experimental group compared to the control group (*p* < 0.01) (Figure 7).

### 2.6. Recombinant Cap Protein Can Induce Cellular Immune Response in Mice after Immunization

Cytokines in the serum samples of mice at 6 weeks after immunization were detected, and the concentrations of IL-2, IL-4 and IF-γ in the experimental group were significantly higher than those in the control group (*p* < 0.01) (Figure 8).

### 2.7. Optimized I-ELISA Method

The optimized best response procedures: envelope antigen concentration 1.0 μg/mL, 37 °C for 2 h. Next, 30 g/L BSA PBS was used as a sealing solution at 37 °C for 1 h. The serum was diluted at 1: 200 and reacted at 37 °C for 45 min. An enzyme-conjugate secondary antibody was diluted at 1:10,000 and reacted at 37 °C for 30 min. The chromogenic substrate reacted at 37 °C without light for 15 min (Table 1, Table 2 and Table 3).

### 2.8. Establishment of Negative and Positive Determination Criteria for I-ELISA

The 31 negative sera were detected by the optimized ELISA method. The data statistics, the average serum of OD_450_
$\overset{-}{x}$ = 0.206 and standard deviation SD = 0.026. Therefore, with the OD_450_ ≤ 0.258 of the test serum to be tested, it was considered as negative. Furthermore, serum OD_450_ ≥ 0.284 was determined to be positive. When 0.258 < OD_450_ < 0.284, the serum to be tested was considered as suspicious and needed to be tested again (Figure 9).

### 2.9. Sensitivity Determination

When the positive serum was diluted from 1:20, it could still be identified as positive when the serum was diluted to 1:2560, indicating that the established method had high sensitivity (Table 4).

### 2.10. Specificity Determination

The established I-ELISA was used to test PCV3 positive sera, PCV3 negative sera, PCV2 (porcine circovirus type 2), PRRSV (porcine reproductive and respiratory syndrome virus), CSFV (porcine fever virus), PPV (porcine microvirus), PRV (pseudorabies virus), JEV (Japanese encephalitis virus), PEDV (porcine epidemic diarrhea virus) and TGEV (porcine infectious gastroenteritis virus) positive sera. The findings demonstrated the high specificity of the established approach, with only PCV3 positive sera being positive and the remainder being negative (Table 5).

### 2.11. Repeatability Determination

The serum samples were detected using the established I-ELISA method, and the coefficient of variation was estimated to be 2.07–4.23% for intra-batch and 2.14–3.88% for inter-batch. This demonstrated that the established I-ELISA approach delivered reliable detection results and acceptable repeatability (Table 6 and Table 7).

### 2.12. Antibody Detection Using the I-ELISA

The established I-ELISA method was used to detect 120 samples with suspected PCV3 infection stored in the laboratory. Only 30 samples out of 120 tested positive using conventional PCR, with a detection rate of 25%, while 51 samples tested positive using I-ELISA, with a detection rate of 42.5%. The findings demonstrated that the I-ELISA approach was sensitive and suitable for clinical serum sample detection (Table 8).

## 3. Discussion

In recent years, porcine circovirus type 3 has been widespread in the world, and the infection has been increasing year by year, which brings huge economic losses to the pig industry. So far, there is no effective treatment or vaccine on the market at home or abroad. As the only structural protein of the virus, PCV3 Cap is highly sequentially conserved. The study of the structure and function of PCV3 Cap is of great significance for the subsequent development of sensitive and efficient detection kits for PCV3 and corresponding subunit vaccines.

The purpose of this study is to establish the expression system of foreign genes of the recombinant Cap protein. Currently, there are two major systems in the field of recombinant proteins. Mammalian cells and *Escherichia coli* are the most common eukaryotic and prokaryotic expression systems. The two expression systems have advantages and disadvantages. Mammalian cell expression systems have the function of eukaryotic post-translational processing and modification, and the foreign proteins expressed are more similar to natural proteins. The disadvantage is that the cost is expensive, the expression level is low and sometimes the protein cannot be secreted. The expression system of *Escherichia coli* has the advantages of high foreign protein expression, clear genetic background, clear biochemical characteristics, simple operation and relatively low cost of culture, so it has become the preferred expression system of engineering vaccines.

As the n-terminal nuclear localization signal sequence of the Cap protein contains abundant arginine residues and an *Escherichia coli* rare codon, a complete Cap protein cannot be expressed by the prokaryotic expression system, or the protein expression level is too low [11]. Therefore, the expression of the Cap protein requires codon optimization of the n-terminal nuclear localization signal sequence or truncation of the region. In this experiment, the absence of a nuclear localization signal sequence was adopted to construct the expression vector. In this study, we first constructed the recombinant plasmid pET-28a-Cap, selected *Escherichia coli* as the gene expression strain and successfully expressed the recombinant protein Cap in *Escherichia coli* Rosetta (DE3). However, the Cap gene is a heterogenic gene for *Escherichia coli*, and the codon preference of *Escherichia coli*, induction concentration, induction temperature and other factors will lead to abnormal folding of the expressed protein, resulting in the formation of inclusion bodies. Western blotting and SDS analysis confirmed the successful expression of the recombinant Cap protein in the precipitation. The band size was about 25 KD. The recombinant Cap protein was successfully assembled into VLPs with a size of about 20 nm by electron microscopy. The size of the recombinant Cap protein-assembled VLPs expressed in this experiment was different from the size of the VLPs expressed by Wang [11], which might be because different Cap protein sizes, different buffers and other factors would affect the size of VLPs. At the same time, we found that the number of Cap proteins assembled into VLPs expressed in this study was lower compared to Wang. We speculate that this may be due to the fact that PCV3, similar to PCV2, cannot form stable VLPs in a solution with a certain number of amino acids truncated at the N terminus of NLS [12]. Our results were consistent with those of Ji [13], which expressed the recombinant Cap protein in truncated form and assembled fewer VLPs.

In this study, the recombinant Cap protein was immunized in mice, and the test results confirmed that the recombinant Cap protein had certain immunogenicity and could induce higher levels of antibodies with the increase in the number of immunizations. The antibody dilution can reach more than 200,000 times. The results of this study further confirmed that the recombinant Cap protein can be assembled into VLPs, and VLPs can stimulate a strong humoral immune response in mice. This experiment was similar to the antibody titer of Ji. [13] after immunizing rabbits with a recombinant Cap protein expressed in prokaryotes, which reached more than 200,000 times. Cellular immunity is an important antiviral mechanism. It also offers broader protection when T-cell-mediated immune responses are combined with humoral immune responses, and CD4^+^ and CD8^+^ T cells are needed for antiviral effects. CD4^+^T cells can differentiate into Thl and Th2 subsets. Thl cells mainly mediate cellular immune responses and secrete cytokines such as IFN- and IL-2. Th2 cells play a very important role in humoral immunity, mainly by secreting IL-4, IL-5 and other cytokines. CD8^+^T cells play an important role in recognizing and presenting antigens and can specifically kill virus-infected cells [14]. The results of this study showed that compared with the control group, CD3^+^CD4^+^ and CD3^+^CD8^+^ cells in mice immunized with the recombinant Cap protein were significantly increased (*p* < 0.05, *p* < 0.01), which may help to enhance the protective immune response of the body. The expressions of cytokines IL-4, IL-2 and IFN-γ were significantly higher than those of the control group (*p* < 0.01), suggesting that the recombinant protein-immunized mice could induce the antigen-specific responses of Th1 and Th2. These results indicated that the recombinant Cap protein could induce better cellular and humoral immune responses in mice.

Since the clinical symptoms caused by PCV3 and PCV2 are very similar, the diagnosis of PCV3 should be based on a laboratory differential diagnosis. Currently, the laboratory differential diagnosis methods for PCV3 mainly include molecular biological diagnosis and serological diagnosis, among which PCR, microdrop digital PCR [15], quantitative fluorescence PCR [16], loop-mediated isothermal amplification (LAMP) [17] and other molecular biological methods can detect specific nucleotide sequences of PCV3. This method has the characteristics of high sensitivity, strong specificity and good accuracy. It has been widely used in the clinical diagnosis of PCV3 and PCV2 infection in pig farms and has achieved a good application effect. However, molecular biological methods have a high detection cost and a limited detection amount, so they are not suitable for large quantities of samples or large-scale epidemiological monitoring. Serological diagnostic methods such as ELISA have simple operations, a short detection cycle, low detection costs and high throughput, which make them especially suitable for the serological monitoring of large quantities of samples or large areas.

In this study, the *E. coli* prokaryotic expression system was selected for in vitro expression of the Cap protein encoded by the OPR2 gene in PCV3, which is also the only capsid protein of PCV3. The results of subsequent experiments confirmed that the truncated Cap protein was expressed in large amounts, easily purified and could be assembled into VLPs in vitro with good reactivity. The VLPs are more suitable for the development of serological diagnostic tests because they mimic the structure of a virus with repeated surface antigenic epitopes in an appropriate conformation compared to monomeric viral candidate antigens. VLP-based enzyme-linked immunosorbent assays (ELISA) are widely used to measure antibodies or neutralizing epitopes. In the present study, we established a VLP-based ELISA antibody detection method. The I-ELISA established using a coating antigen differs from the method established by Wang [11], in that we chose a truncated expression of the recombinant Cap protein, whereas Wang’s was full-length optimized. This was also different from Deng [18], in that although they were truncated expressions, they were not assembled into VLPs. In contrast to the ELISA method established by Zhang [19] using a baculovirus expression system, it did not assemble into VLPs in vitro. The method established in this study had a minimum detection titer of 1:2560 for a PCV3 positive serum, and the coefficient of variation in both intra-batch and inter-batch repeatability tests was less than 5%, showing good sensitivity and repeatability.

As a new porcine virus, positive results need to be detected by PCR or ELISA. The ELISA has the benefits of simple operation and high throughput, and it can quickly ascertain whether pigs are PCV3-infected. The ELISA technique used in this work demonstrated good sensitivity to positive sera and no cross-reactivity with other pathogens, such as porcine circovirus type 2 (PCV2). The ELISA method, used to identify antibodies in 120 suspected clinical samples stored in the laboratory, demonstrated greater sensitivity compared to traditional PCR, indicating that the method is appropriate for epidemiological research.

## 4. Materials and Methods

### 4.1. Plasmid Construction

The PCV3 Cap gene sequence (GenBank: MN310686.1) was obtained from NCBI, and *Xho* I and *BamH* I restriction sites were inserted into the 3′ end and 5′ end, respectively, by truncating the 27 aa N-terminal, and His tag sequence was introduced into the 3′ end. The gene sequence was then cloned into the pET-28a vector (Jin Weizhi Company, Tianjin, China). The Jin Weizhi Company (Jin Weizhi Company, Tianjin, China) was commissioned to prepare the plasmid. The synthesized plasmid was propagated in an ampicillin (Merck, Shanghai, China)-resistant LB medium (TransGen Biotech, Beijing, China), and the endotoxin-free plasmid was extracted through the plasmid maxi preparation kit (Beyotime, Shanghai, China). It was then sent to the company for sequencing and the correct sequencing plasmid was received.

### 4.2. Eukaryotic Expression of Recombinant Cap Protein

The correctly sequenced plasmid, called the pET-28a-Cap plasmid, was transformed into *E. coli* BL21 (DE3) (Takara, Beijing, China) in 50 mL LB medium (TransGen Biotech, Beijing, China) containing 50 μg of kanamycin (Merck, Shanghai, China) and shaken overnight at 37 °C, 220 rpm. The medium was cultured in 200 mL fresh LB medium at a ratio of 1:10 and shaken at 220 rpm at 37 °C for 6–8 h. When the OD_600_ was 0.6–0.8, the final concentration of isopropyl-β-D-galactoside (IPTG) (Beyotime, Shanghai, China) was added for 1 mM, and shaken at 220 rpm at 22 ℃ for 20 h. The medium was harvested by centrifugation at 8000 rpm at 4 °C for 30 min. The precipitate was re-suspended in a PBS buffer solution (137 mM NaCl, 2 mM KH_2_PO_4_, 25 mM Na_2_HPO_4_, 2 mM KCl) and treated with ultrasonic treatment on ice. Finally, the supernatant and precipitate were harvested by centrifugation at 10,000 rpm for 30 min at 4 ℃. The precipitate was re-suspended with the same volume of PBS. The expression and solubility of the recombinant Cap protein were analyzed by sodium dodecyl sulfate polyacrylamide gel electrophoresis (SDS-PAGE) and Western blotting. The primary antibody was PCV3 porcine positive serum, and the secondary antibody was goat anti-porcine horseradish peroxidase IgG (1:1000) (Kingsray Biotechnology, Nanjing, China). The specific operation experiment was as follows: SDS loading buffer (Beyotime, Shanghai, China) was added to the supernatant to be analyzed and boiled in boiling water for 10 min to denature the protein. Proteins were then isolated on 12% polyacrylamide gel (Boao Yijie technology, Beijing, China). The separation gel was stained with Coomassie Bright Blue R-250 (Beyotime, Shanghai, China). At the same time, another separation glue was transferred to the PolyScreen PVDF transfer film (Merck, Shanghai, China). The membranes were treated with a closed solution (PBS containing 5% skim milk) for 1 h, incubated with PCV3 porcine positive serum (1:200) (our laboratory preservation) at room temperature for 2 h, washed with PBS 3 times for 5 min each time, incubated with goat anti-pig IgG horse radish peroxidase conjugate (1:1000) (Kingsray Biotechnology, Nanjing, China) at room temperature for 1 h and washed with PBS 3 times for 5 min each time. The BeyoECL Plus Western blot Chemiluminescence kit (Beyotime, Shanghai, China) was used for color development after treatment. The Tanon5200 protein imaging analysis instrument (Tanon, Shanghai, China) was then used for signal detection. In order to improve the expression level of the recombinant Cap protein, the induction conditions of IPTG were optimized, and the concentrations were 0.8 mM, 1.0 mM and 1.2 mM. Then, purification was performed by nickel ion affinity chromatography (Sango, Shanghai, China). The target proteins collected after purification were dialyzed with 6, 4, 3, 2, 1 and 0 mol/L urea buffers, and each gradient was dialyzed for 4 h with a dialysis bag (Thermo Fisher, Shanghai, China), and then dialysated with dialysate (500 mM NaCl, 2 mM KH_2_PO_4_, 25 mM Na_2_HPO_4_, 5 mM β-Me, 2 mM KCl and 10 mM Tris. PH = 8.5) for overnight dialysis at 4 °C. The protein solution after dialysis was concentrated by an ultrafiltration tube.

### 4.3. Transmission Electron Microscopy (TEM)

The concentrated Cap protein samples were adsorbed onto a carbon-coated copper grid (Zhongjingkeyi, Beijing, China) for 3 min at room temperature. The grating was gently dried with filter paper and stained with 2% phosphotungstic acid (Merck, Shanghai, China) for 40 s. The excess liquid was removed with filter paper and the sample was observed under a transmission electron microscope, JEOL JEM 1400 TEM (JEOL Limited, Tokyo, Japan).

### 4.4. Immunization

Ten six-week-old female BALB/c mice (Northwest A&F University Laboratory Animal Center) without specific pathogens were randomly divided into 2 groups of 5 mice in each group (experimental group and control group). The experimental group mice were vaccinated with 200 μL of vaccine containing 25 μg of recombinant Cap protein emulsified with adjuvant Montanide ISA201 (Seppic, Paris, France) in the thigh muscle and subcutaneously. The vaccine was prepared as follows: by taking 100 μL of recombinant Cap protein containing 25 μg, adding 100 μL of Montanide ISA201 adjuvant according to 1:1 volume, shaking at room temperature for 30 min and then using. The control group was PBS, which was also emulsified according to the volume ratio of 1:1, 100 μL PBS and 100 μL adjuvant. After the initial immunization, vaccinations were given twice 3 weeks apart. After immunization, serum samples were collected at 1, 2, 3, 4, 5 and 6 weeks, respectively. An indirect enzyme-linked immunosorbent assay (I-ELISA) was used to analyze the IgG response level of the recombinant Cap protein specific antibody in serum samples. For each group, all mice were euthanized 6 weeks after immunization. The serum was collected for subsequent test assays.

### 4.5. Indirect Enzyme-Linked Immunosorbent Assay (iELISA)

ELISA plates were coated with the recombinant Cap protein in 100 ng per well and diluted with coated buffer solution overnight at 4 °C. PBST (Merck, Shanghai, China) was washed three times, after reoccupying the closed fluid sealing, 100 μL per well, at room temperature for 1 h. The liquid was drained, dried at room temperature, bagged and set aside. Then, 1000-fold diluted serum for 1–6 weeks and diluted serum for 6 weeks were added to each well (1:100, 1: 1000, 1:10,000, 1:100,000, 1:200,000, 1:400,000), as the primary antibody, 37 ℃ incubator for 1 h, PBST washed the plate 3 times. The second antibody was used: HRP Goat anti-mouse IgG (1:10,000) (Kingsray Biotechnology, Nanjing, China), 37 °C incubator for 1 h and PBST washed the plate 3 times. After that, the TMB color-developing solution (Surmodics, Beijing, China) was added to 100 μL per well for 8 min, and then 50 μL per well was added to stop the solution (2 M H_2_SO_4_) quickly. Then, OD_450_ was measured by Biotek ELx800 spectrophotometer (Biotek, Beijing, China). The experiment was repeated three times. A ratio greater than 2.1 was considered positive.

### 4.6. Assay of T Lymphocyte Subsets in Peripheral Blood

The collected 200 μL of blood by capillary from mice was added to a flow tube (Corning, Shanghai, China) containing heparin sodium and mixed gently to prevent clotting. Antibody incubation (CD3: 1.5 μL, CD8: 1.5 μL, CD4: 0.9 μL) (Biosciences, Brisbane, CA, USA) occurred at room temperature away from light incubation for 30 min. After the antibody incubation, the whole blood was transferred to a new flow tube, 4 mL of red blood cell lysate was added, the blood was lysed for 10 min away from light, centrifuged with a horizontal rotor at 4 °C and 2000 rpm for 5 min and the supernatant was discarded and repeated. The cells were re-suspended with 300 μL of Histopaque sorting solution (Merck, Shanghai, China). An analysis was performed using a flow meter (Beckman CytoFLEX, Shanghai, China). The experiment was repeated three times.

### 4.7. Cytokines Release Assay

Serum samples from mice immunized for 6 weeks were tested using mouse IL-2, IL-4 and IFN-γ ELISA kits (Shanghai Enzyme Linked Biological Company, Shanghai, China) as follows: the plates were removed from aluminum foil bags after 60 min of equilibration at room temperature. Standard wells, blank wells and sample wells were set up, and three wells were operated in each group. Standard wells with different concentrations of 50 μL of the diluted standard were added to each standard well. Sample wells were added with 50 μL of the sample to be tested, and blank wells were added with 50 μL of the sample diluent. After that, 100 μL of horseradish peroxidase (HRP)-labeled detection antibody was added to each well and incubated for 60 min in an incubator at 37 °C. The liquid was discarded, washed three times and after draining, 50 μL of substrate A and B was added to each well and incubated for 15 min in an incubator at 37 °C. Then, 50 μL of termination solution was added to each well, and after 15 min, OD values were measured at 450 nm.

### 4.8. Determination of Optimal Working Concentration of Recombinant Protein and Serum

The purified recombinant Cap proteins were coated on the ELISA plate with a coating solution (1.59 g/L Na_2_CO_3_, 2.93 g/L NaHCO_3_) as 5 dilutions of 10 μg/mL, 5 μg/mL, 2.5 μg/mL, 1.0 μg/mL and 0.5 μg/mL, respectively. In addition, there was 100 μL/well, overnight at 4 °C. The next day, the coating solution was discarded, PBST was washed three times and PBST containing 0.5% bovine serum albumin (Merck, Shanghai, China) was added as 100 μL/well and incubated at 37 °C for 2 h. The pig positive serum and negative serum were diluted 1:50, 1:100, 1:200 and 1:400 times with serum diluent (Merck, Shanghai, China) to determine the optimal working concentrations of antigen and serum as 100 μL/well. It was 37 °C incubator for 1 h, PBST washed the plate 3 times and the second antibody used HRP Goat anti-pig IgG (Kingsray Biotechnology, Nanjing, China), 37 °C incubator for 1 h, PBST washed the plate 3 times. After that, the TMB color-developing solution (Surmodics, Beijing, China) was added to 100 μL per well for 8 min, and then 50 μL per well was added to stop the solution (2 M H_2_SO_4_) quickly. OD_450_ was measured by Biotek ELx800 spectrophotometer (Biotek, Beijing, China).

### 4.9. Determination of Serum Working Time

According to the above optimized conditions, the pig positive serum and negative serum were added to a 96-well ELISA plate with 100 μL/well and treated at 37 °C for 45, 60 and 90 min. The OD_450_ values of each group were compared, and the corresponding time of the maximum P/N value was the optimal working time of the serum.

### 4.10. Determination of Optimal Dilution Ratio and Working Time of Secondary Antibody

According to the above optimized conditions, the secondary antibody HRP Goat anti-pig IgG (Kingsray Biotechnology, Nanjing, China) was diluted 1:5000, 1:10,000, 1:20,000 and 1:40,000 times; added to a 96-well ELISA plate; acted at 37 °C for 30, 45 and 60 min; and the OD_450_ values of each group were compared. The conditions corresponding to the maximum P/N value were used as the best dilution of the secondary antibody and the best working time.

### 4.11. Determination of Cut-Off Value

By using the optimized reaction conditions, with 29 negative serum samples for testing, we calculated the average sample OD_450_ ($\overset{-}{x}$) and standard deviation (SD). According to the principle of statistics, OD_450_ value ≤ $\overset{-}{x}$ + 2SD was judged as negative, OD_450_ value ≥ $\overset{-}{x}$ + 3SD was judged as positive and $\overset{-}{x}$ + 2SD < OD_450_ value < $\overset{-}{x}$ + 3SD was judged as suspicious samples.

### 4.12. Sensitivity Assay

The PCV3 positive sera were treated with 1:20, 1:40, 1:80, 1:160, 1:320, 1:640, 1:1280, 1:2560, 1:5120 and 1:10,240 times dilution. The established indirect ELISA method was used to detect each dilution, determine the lowest detectable titer and determine the sensitivity of the indirect ELISA method.

### 4.13. Cross-Reactivity Assay

The positive serum was of classical swine fever virus (CSFV), porcine circovirus 2 (PCV2), Japanese encephalitis virus, (JEV), Pseudorabies virus (PRV), porcine reproductive and respiratory syndrome virus (PRRSV), transmissible gastroenteritis of pigs virus (TGEV) and porcine epidemic diarrhea virus (PEDV). PCV3 positive and negative sera were used as the control, and the established indirect ELISA method was used to cross-test.

### 4.14. Reproducibility Assay

Intra- and inter-batch reproducibility assays: Seven serum samples of PCV3 were tested at four different time points using recombinant Cap protein-coated ELISA plates prepared from the same and different batches, and the coefficients of variation were calculated to determine the intra- and inter-batch reproducibility of the I-ELISA method.

### 4.15. Antibody Detection Using the PCV3 VLP-ELISA

The established I-ELISA method was used to detect 120 suspected PCV3 infection samples stored in the laboratory, so as to determine the clinical application of the I-ELISA method established in this study.

### 4.16. Statistical Analysis

A one-way ANOVA significance analysis was performed using Graphpad Prism software 7.0; significance was defined as a *p* value less than 0.05. A *p* value greater than 0.05 was not significant, and a *p* value less than 0.01 was considered highly significant.

## 5. Conclusions

In conclusion, this study revealed for the first time that the truncated form of the PCV3 recombinant Cap protein was successfully produced in *E. coli* and assembled into VLPs in vitro. Additionally, an I-ELISA approach was created and used to identify clinical antibodies specific to the PCV3 serum. The ELISA based on VLPs is an effective method for tracking PCV3 prevalence, since it is highly specific, sensitive and reproducible. The experiment is at a preliminary stage, and further research is required to investigate how to form stable and non-degradable VLPs in a solution and to improve the assembly efficiency of VLPs.

## Figures and Tables

**Figure 1 ijms-24-10377-f001:**
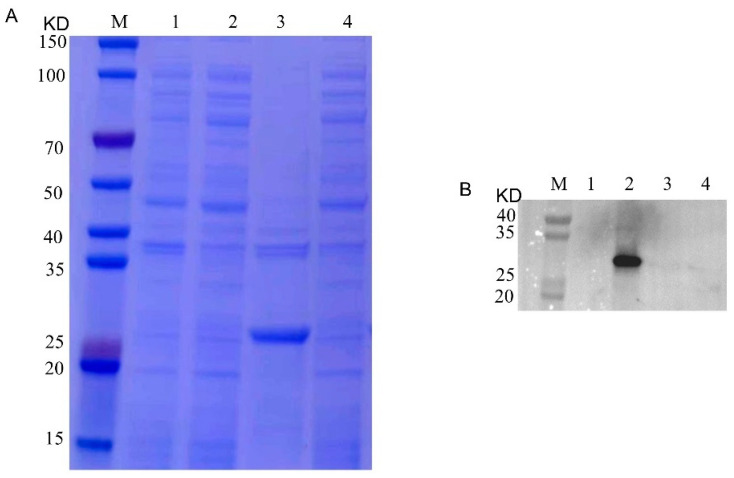
SDS-PAGE and Western blotting analysis of prokaryotic expression of recombinant Cap protein. (**A**): M, 1, 2, 3 and 4 represent the protein marker, empty carrier precipitation, empty carrier supernatant, recombinant Cap protein precipitation and recombinant Cap protein supernatant, respectively. (**B**): M, 1, 2, 3 and 4 represent protein marker, recombinant Cap protein supernatant, recombinant Cap protein precipitation, empty carrier supernatant and empty carrier precipitation, respectively.

**Figure 2 ijms-24-10377-f002:**
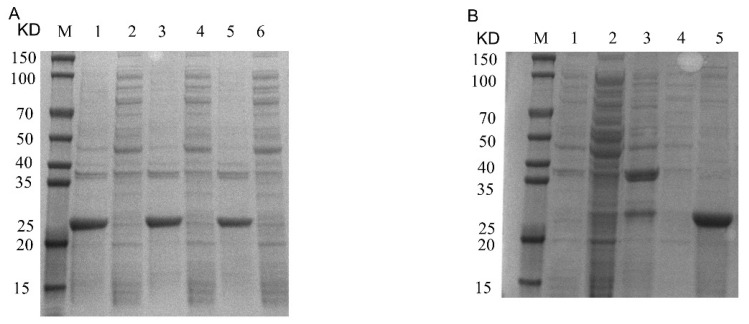
Recombinant Cap protein prokaryotic expression of IPTG-induced concentration and protein purification SDS-PAGE. (**A**): M, 1, 2, 3, 4, 5 and 6 represent protein marker, 1.2 mM precipitation, 1.2 mM supernatant, 1.0 mM precipitation, 1.0 mM supernatant, 0.8 mM precipitation and 0.8 mM supernatant, respectively. (**B**): M, 1, 2, 3, 4 and 5 represent flow-through, unpurified protein, 50 mM wash first time, 50 mM wash second time and elution, respectively.

**Figure 3 ijms-24-10377-f003:**
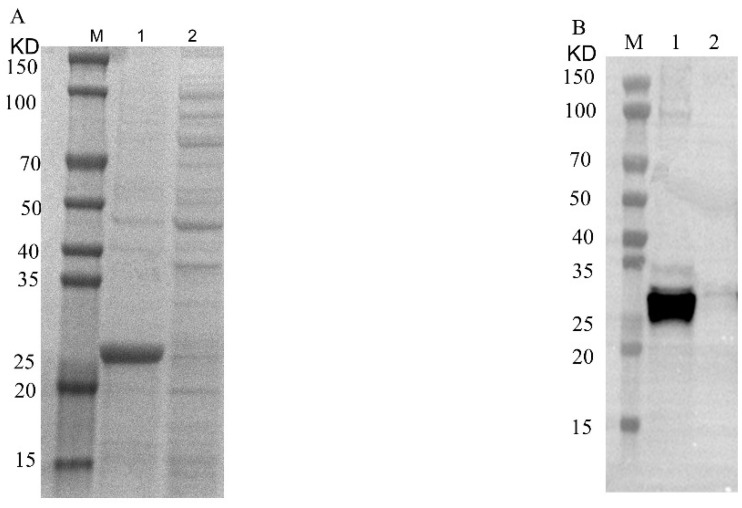
SDS-PAGE and Western blotting analysis of recombinant Cap protein after concentration. (**A**): M, 1 and 2 represent protein marker, concentrated recombinant Cap protein and flow-through, respectively. (**B**): M, 1 and 2 represent protein marker, concentrated recombinant Cap protein and flow-through, respectively.

**Figure 4 ijms-24-10377-f004:**
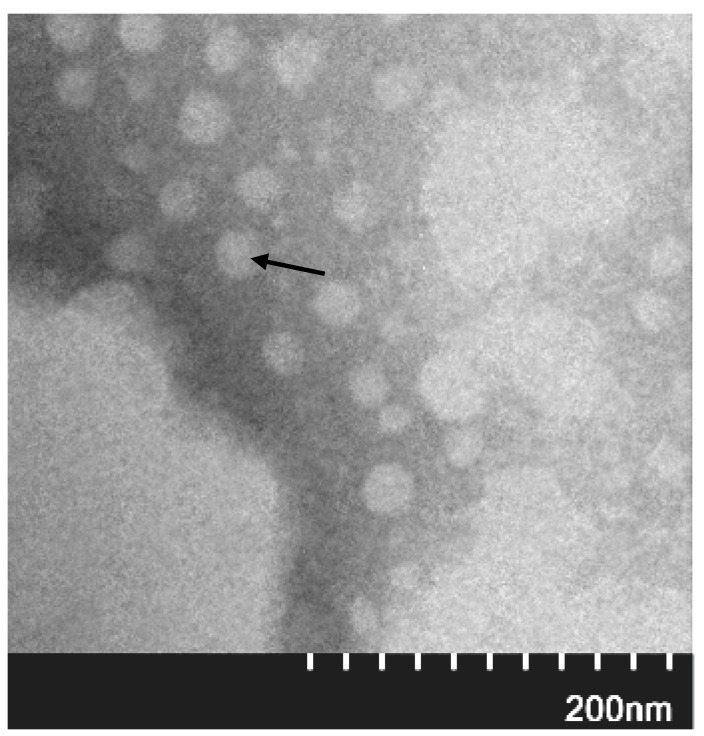
VLP analysis of recombinant Cap protein assembly. The black arrow indicates the assembled VLP.

**Figure 5 ijms-24-10377-f005:**
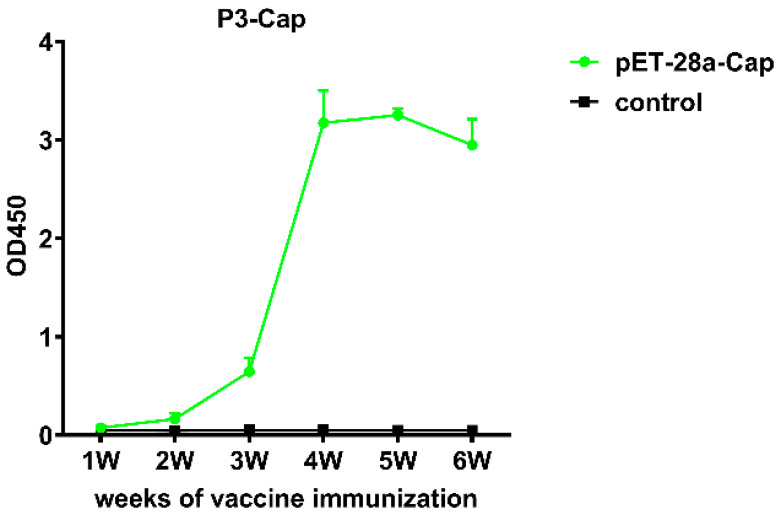
Results of antibody immune response in mice immunized with recombinant Cap protein.

**Figure 6 ijms-24-10377-f006:**
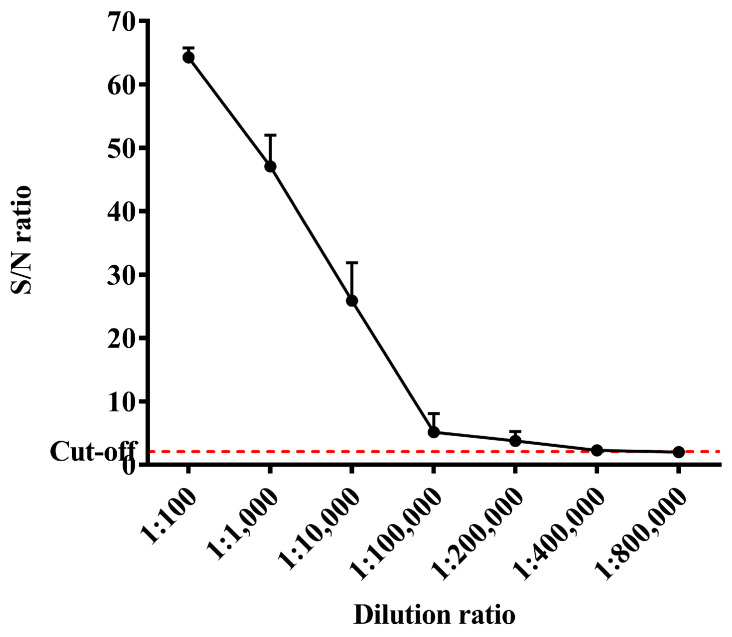
Results of antibody response levels in mice immunized with recombinant Cap protein.

**Figure 7 ijms-24-10377-f007:**
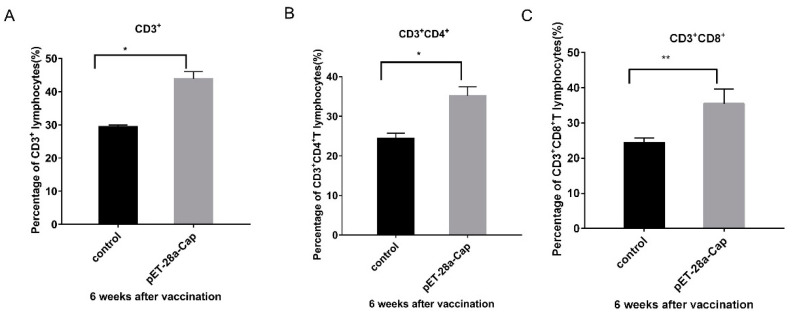
Results of T cell proliferation in peripheral blood samples of the recombinant Cap protein immunization group and the control group. (**A**) is the result of CD3^+^ T cell analysis; (**B**) is CD3^+^ CD4^+^ T cell analysis; (**C**) is the result of CD3^+^ CD8^+^ T cell analysis; * *p* < 0.05; ** *p* < 0.01.

**Figure 8 ijms-24-10377-f008:**
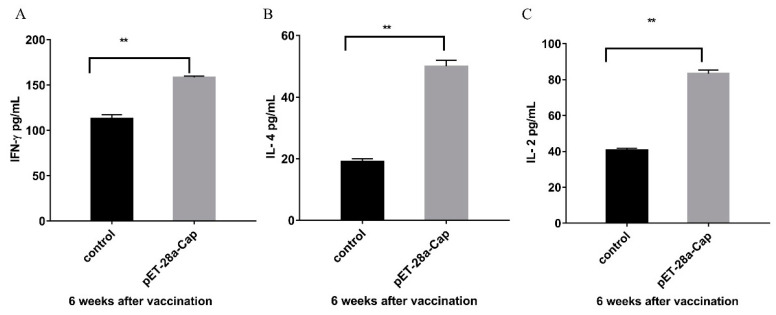
Results of serum cytokines in the recombinant protein immune group and the control group. (**A**) is the result of IF-γ analysis; (**B**) is the IL-4 analysis result; (**C**) is the IL-2 analysis result. ** *p* < 0.01.

**Figure 9 ijms-24-10377-f009:**
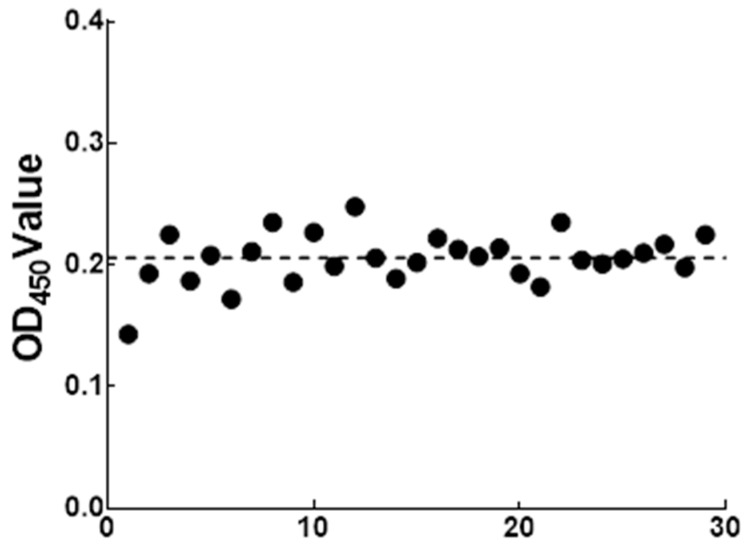
Cut-off value definition of the indirect ELISA method.

**Table 1 ijms-24-10377-t001:** Determination of optimal concentration of antigen and dilution of serum.

Serum Dilution	Coating Concentration of Antigen/μg/mL							
				10	5	2.5	1.0	0.5
1:50			P	1.322	1.30	1.275	1.25	1.024
			N	0.354	0.29	0.283	0.267	0.243
			P/N	3.734	4.482	4.505	4.681	4.213
1:100			P	1.277	1.155	1.20	1.188	0.980
			N	0.312	0.281	0.270	0.255	0.221
			P/N	4.092	4.110	4.445	4.658	4.434
1:200			P	1.104	1.075	1.087	1.078	0.904
			N	0.214	0.201	0.182	0.177	0.165
			P/N	5.159	5.348	5.972	6.090	5.478
1:400			P	1.011	1.004	1.002	0.985	0.833
			N	0.184	0.179	0.175	0.165	0.152
			P/N	5.494	5.608	5.725	5.969	5.480

**Table 2 ijms-24-10377-t002:** Determination of optimal serum reaction time.

Reaction Time of Serum/min	OD450 Value			P/N
		P	N	
45		1.203	0.175	6.874
60		1.321	0.231	5.71
90		1.470	0.286	5.139

**Table 3 ijms-24-10377-t003:** Determination of optimal dilution and reaction time of HRP-Rabbit anti pig.

Dilution Of HRP-Rabbit Anti Pig	Reaction Time/Min	OD450 Value		P/N
		P	N	
1:5000	30	1.422	0.325	4.375
	60	1.530	0.541	2.828
	90	1.855	0.610	3.040
1:10,000	30	1.084	0.182	5.956
	60	1.401	0.252	5.559
	90	1.543	0.342	4.511
1:20,000	30	0.672	0.133	5.052
	60	0.955	0.182	5.247
	90	1.146	0.221	5.185
1:40,000	30	0.133	0.070	1.900
	60	0.201	0.093	2.161
	90	0.259	0.105	2.466

**Table 4 ijms-24-10377-t004:** Sensitivity test.

Radio of serum dilution	20	40	80	160	320	640	1280	2560	5120	10,240
OD_450_ value	1.542	1.434	1.212	1.103	0.945	0.735	0.582	0.346	0.280	0.103
Result	+	+	+	+	+	+	+	+	−	−

“+”: positive “−“: negative.

**Table 5 ijms-24-10377-t005:** Results of specificity test.

Serum Samples	OD_450_ Value	Result
PRRSV	0.102	−
CSFV	0.131	−
PPV	0.042	−
PRV	0.065	−
JEV	0.083	−
PEDV	0.113	−
TGEV	0.092	−
PCV2	0.076	−
PCV3 positive	1.206	+
PCV3 negative	0.135	−

“+”: positive “−“: negative.

**Table 6 ijms-24-10377-t006:** Repeatability test results for inter-batch.

Number of Samples	OD_450_ Values at 4 Different Time Points				Average Value	Standard Deviation	Coefficient of Variation/%
1	1.197	1.257	1.155	1.249	1.214	0.035	2.87%
2	1.025	1.035	1.027	1.078	1.041	0.021	2.07%
3	0.725	0.715	0.751	0.713	0.726	0.015	2.08%
4	0.586	0.537	0.566	0.581	0.567	0.019	3.37%
5	0.345	0.365	0.341	0.361	0.353	0.015	4.17%
6	0.524	0.521	0.490	0.501	0.509	0.014	2.75%
7	0.332	0.345	0.378	0.364	0.355	0.015	4.23%

**Table 7 ijms-24-10377-t007:** Repeatability test results for intra-batch.

Number of Samples	OD_450_ Values at 4 Different Time Points				Average Value	Standard Deviation	Coefficient of Variation/%
1	1.102	1.153	1.166	1.145	1.141	0.028	2.53%
2	1.156	1.145	1.224	1.207	1.183	0.036	3.04%
3	0.987	0.925	0.972	0.934	0.954	0.037	3.88%
4	0.856	0.876	0.852	0.831	0.848	0.022	2.59%
5	0.785	0.746	0.768	0.734	0.758	0.019	2.50%
6	0.658	0.634	0.659	0.664	0.653	0.014	2.14%
7	0.552	0.563	0.582	0.536	0.558	0.018	3.22%

**Table 8 ijms-24-10377-t008:** Results of detection by ELISA and conventional PCR for clinical samples.

	The Conventional PCR		
	Positive	Negative	Total
I-ELISA			
Positive	30	21	51
Negative	0	69	69
Total	30	90	120

## Data Availability

The data presented in this study are available on request from the corresponding author. The data are not publicly available due to intellectual property considerations.

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
