# Peer review of "Immunogenicity Analysis of PCV3 Recombinant Capsid Protein Virus-like Particles and Their Application in Antibodies Detection"

_ijms, 2023, doi:10.3390/ijms241210377_

Round 1

Reviewer 1 Report

The study is well designed, the results are clear, conclusions are moderate, correct. I  have only some critical points regarding to the content and some minor advices to improve the presentation of the results.

1. Since the topic of PCV3 is very 'hot', there are several novel publication published day by day. Hence the list of references should be made a little bit up to date. Please implement one or two recent article about  PCV3!

In line 8 you mention that PCV3 couses 'huge economic losses'. It may be disputable, please mention some arguments with references pros and cons in the Introduction.

Please extend the last paragraph of the Introduction with some remarks on IFN-gamma and other cytokines and the recent knowledge on their role in PCV3 infections.

2.  Improve the quality of gel photos (contrast, brightness)!

Fig 4: Expand the captation. All of the figures should be understandable "standing alone". Improve the photo quality!

Fig. 5. 6. and 7.: More details in captation. Don't use "Analysis of..." but "Results of the xxxxx-specific I-ELISA on ...."! Enlarge the Figures, at least the texts (legends)!

Line 237: Insert the manufacturer of pET-28a vector. Add the city / country as well! similarly in the case of Jin Weizhi Company.

Line 238: use ampicillin instead of Amp.

line 266: Write more details of the emulsion! What was the aqueous phase, how was it actually made?

Section 4.4,  What was the 'control group' how was it treated? Any mock vaccination?

English is very good, however try to fragment the long sentences in M&M  section. It would help the reader to understand the process step by step.

In sentences listing the different circumstances of alternative treatments, use  ‘respectively’! Like in section 4.10: "... for 30, 45 and 60 min, respectively".

Author Response

Reviewer: 1

The study is well designed, the results are clear, conclusions are moderate, correct. I  have only some critical points regarding to the content and some minor advices to improve the presentation of the results.

  1. Since the topic of PCV3 is very 'hot', there are several novel publication published day by day. Hence the list of references should be made a little bit up to date. Please implement one or two recent article about  PCV3!

Response: Thank you very much for your advice. I have inserted one recent article about  PCV3. “10.     Deng, H.; Zhu, S.; Zhu, L.; Jian, Z.; Zhou, Y.; Li, F.; Deng, L.; Deng, J.; Deng, Y.; Lai, S. Histopathological Changes and Inflammatory Response in Specific Pathogen-Free (SPF) with Porcine Circovirus Type 3 Infection. Animals 2023, 13 (3), 530.”

In line 8 you mention that PCV3 couses 'huge economic losses'. It may be disputable, please mention some arguments with references pros and cons in the Introduction.

Response: Thank you very much for your advice. I think the phrase "huge economic losses" is really disputable. Porcine circovirus type 3 is a newly emerging pathogen, which is widely prevalent at home and abroad, but the current research on its pathogenicity is not clear enough. So I decided to delete the word "huge" to avoid controversy.

Please extend the last paragraph of the Introduction with some remarks on IFN-gamma and other cytokines and the recent knowledge on their role in PCV3 infections.

Response: Thank you very much for your advice. I have added the cytokines on he recent knowledge on their role in PCV3 infections. As follows “Pro-inflammatory cytokines play a key role in the development and maintenance of inflammation and can lead to multiple organ damage. Previous studies have shown that PCV3 infection up-regulate pro-inflammatory cytokines in pigs[9], Moreover, the PCV3-mediated clinical symptoms of illness and tissue damage may be caused by the high levels of pro-inflammatory cytokines. Recent studies have also confirmed that PCV3 infection does cause inflammatory response, in which IFN-γ, IL-6, IL-8, TNF-α and other cytokines are significantly increased after infection[10].”. Please see introduction the yellow part.

  1. Improve the quality of gel photos (contrast, brightness)!

Response: Thank you very much for your advice. I have corrected it.

Fig 4: Expand the captation. All of the figures should be understandable "standing alone". Improve the photo quality!

Response: Thank you very much for your advice. I have corrected it. Please see Fig.4.

Fig. 5. 6. and 7.: More details in captation. Don't use "Analysis of..." but "Results of the xxxxx-specific I-ELISA on ...."! Enlarge the Figures, at least the texts (legends)!

Response: Thank you very much for your advice. I have corrected "Analysis of..."to"Results of” according to your suggestion. I also enlarged the chart and text. Please see Fig.5.6.7.

Line 237: Insert the manufacturer of pET-28a vector. Add the city / country as well! similarly in the case of Jin Weizhi Company.

Response: Thank you very much for your advice. I have inserted the manufacturer of pET-28a vector. “(Jin Weizhi Company, Tianjin, China)”. Please see M&M 4.1 the yellow part .

Line 238: use ampicillin instead of Amp.

Response: Thank you very much for your advice. I have corrected "Amp"to" ampicillin” according to your suggestion.

line 266: Write more details of the emulsion! What was the aqueous phase, how was it actually made?

Response: Thank you very much for your advice. I have added a detailed vaccine emulsification process. as follows “The vaccine was prepared as follows:Taking 100 mL of recombinant Cap protein containing 25 mg, adding 100 mL of Montanide ISA201 adjuvant according to 1:1 volume, shaking at room temperature for 30 minutes, and then use.”

Section 4.4,  What was the 'control group' how was it treated? Any mock vaccination?

 Response: Thank you very much for your advice. I have added the control group. The control group was inoculated in the same way as the vaccine group

The control group was prepared as follows “The control group was PBS, which was also emulsified according to the volume ratio of 1:1, 100 mL PBS and 100 mL adjuvant.

Reviewer 2 Report

The objective of this paper was to identify immunogenicity of PCV3 recombinant protein. Currently, well known PCV2 virus is still denengerous pathogen, but we have vaccine to protect pigs. Porcine circovirus type 3 is a newly emerging pathogen, but lack of commercially available vaccine. Therefore, the of recombinant capsid protein is a promising tool to prevent and control PCV3 infection.

Major comments:

In Introduction lack of information about PCV2 vaccines and information about usefulness or uselessness in case of PCV3 infection.

Line 250 - The primary antibody was PCV3 porcine positive serum. How Authors detect PCV3 positive sample and in which type of animal (piglets, sow)?

Line 264 - Ten female BALB/c mice of six weeks …without specific pathogens. What is mean specific pathogens?

Line 268, 292 – How serum samples were collected from mice? How much blood was taken?

Line 284 – For T limphocyte assay Authors collected 200 µL of blood from the same mice from which serum was taken? Blood by capillary from mice, but from which place?

Line 283 – 289 – Authors labeled cell with antibodies anti: CD3, CD8 and CD4 and sorted. Is primery qntibodies were directly labeled with fluorophors or Authors used secondary labeled antibodies? Moreover lack of name of sorting solution and instruments model and producer.

Line 290 - Cytokines Release Assay and 302 - Determination of optimal working concentration of recombinant protein and serum, 312 - Determination of serum working time –  Descriptoion of all methods is unclear and has methodological mistakes.

Line 304 - incubated for 2 h at 37 °C and overnight at 4℃ - simultaneously for samleps?

SDS-PAGE and Western blotting figures are not good quality.

Minor comments:

Line 237 – lack of pET-28a vector provider

Line 238 - lack of  Amp resistant LB medium producer; How plasmid was extracted?

Line 243 - medium containing 50 mg of kanamycin - lack of  medium name, producer and volume, lack of kanamycin producer

Line 251 – How Authors detect signal from secondary antibodies? Please add details, reagents and instruments.

Line 253 – lack of ion affinity chromatography instrument/system producer

Line 259 - lack of carbon-coated copper grid producer

Line 254 - lack of dialysys  bag producer

Line 260 - lack of phosphotungstic acid producer

Line 256 - overnight dialysis was in 4°C?

Line 266 - lack of adjuvant Montanide ISA201 provider

Line 273 – 281 – lack of PBST, TMB producer, and spectrophotometer name and producer

Line 284 - lack of  flow tube producer

Line 287 - 400 g should be calculate to rpm

Line 243 - 37℃, 220 rpm

Line 244 – 37℃ for 6-8 hours

Line 245 - China) was

Line 293 - China) as follows

Line 295 - Standard Wells with different concentrations of 50 µL of what?

Line 303 - The purified recombinant Cap proteins were coated on?

Line 305, 306, 309 - lack of  bovine serum albumin, serum diluent, TMB, stop solution producers and spectrophotometer name and producer

Line 318 - lack of  secondary antibody producer

Lin 319 - OD450

Line 354 - in vitro

Author Response

The objective of this paper was to identify immunogenicity of PCV3 recombinant protein. Currently, well known PCV2 virus is still denengerous pathogen, but we have vaccine to protect pigs. Porcine circovirus type 3 is a newly emerging pathogen, but lack of commercially available vaccine. Therefore, the of recombinant capsid protein is a promising tool to prevent and control PCV3 infection.

Major comments:

In Introduction lack of information about PCV2 vaccines and information about usefulness or uselessness in case of PCV3 infection.

 Response: Thank you very much for your advice. I have added the information. As follows“The genome of PCV3 has been sequenced, but the homology with other porcine circoviruses is low, and the sequence homology with PCV2 is only 37% to 40%. There have been no reports of cross-protection against PCV3 with commercially available PCV2 vaccine.”Please see introduction the yellow part.

Line 250 - The primary antibody was PCV3 porcine positive serum. How Authors detect PCV3 positive sample and in which type of animal (piglets, sow)?

 Response: PCR and ELISA were used to detect PCV3 positive sample. The PCV3 positive sample was from piglets.

Line 264 - Ten female BALB/c mice of six weeks …without specific pathogens. What is mean specific pathogens?

Response: It refers to animals that do not carry specific pathogens and parasites that may interfere with experiments, and generally refers to healthy animals without infectious diseases. It is the most widely used experimental animal at home and abroad.

Line 268, 292 – How serum samples were collected from mice? How much blood was taken?

Response: We used glass capillaries to collect blood through orbital veins. 200 ml of blood was collected for subsequent ELISA detection.

Line 284 – For T limphocyte assay Authors collected 200 µL of blood from the same mice from which serum was taken? Blood by capillary from mice, but from which place?

Response: Not the serum. We remove the red blood cells from the blood for subsequent T limphocyte assay. We used glass capillaries to collect blood through orbital veins from mice.

Line 283 – 289 – Authors labeled cell with antibodies anti: CD3, CD8 and CD4 and sorted. Is primery qntibodies were directly labeled with fluorophors or Authors used secondary labeled antibodies? Moreover lack of name of sorting solution and instruments model and producer.

Response: Thank you very much for your advice. CD3, CD8 and CD4  are directly labeled with fluorophors. I have added the name of sorting solution and instruments model and producer. “Histopaque sorting solution(Merck, Shanghai, China).” “flow meter(Beckman CytoFLEX, Shanghai, China).” Please see M&M 4.6 the yellow part .

Line 290 - Cytokines Release Assay and 302 - Determination of optimal working concentration of recombinant protein and serum, 312 - Determination of serum working time –  Descriptoion of all methods is unclear and has methodological mistakes.

Response: Thank you very much for your advice.The Cytokines Release Assay is strictly operated in accordance with the manufacturer's instructions (Shanghai Enzyme Linked Biological Company, Shanghai, China).

I have made some corrections for "Determination of optimal working concentration of recombinant protein and serum". The word "antigen" has been changed to” coating solution”. Please see M&M 4.8 the yellow part.

I have made some corrections for "Determination of serum working time”. The word " serum was " has been changed to” The pig positive serum and negative serum were were”. Please see M&M 4.9 the yellow part

Line 304 - incubated for 2 h at 37 °C and overnight at 4℃ - simultaneously for samleps?

Response: I have deleted the "incubated for 2 h at 37 °C and”. incubated only overnight at 4℃ for samleps. Please see M&M 4.8 the yellow part.

SDS-PAGE and Western blotting figures are not good quality.

Response: Thank you very much for your advice. I have corrected it.

Minor comments:

Line 237 – lack of pET-28a vector provider

Response: Thank you very much for your advice. I have inserted the manufacturer of pET-28a vector. “(Jin Weizhi Company, Tianjin, China)”. Please see M&M 4.1 the yellow part.

Line 238 - lack of  Amp resistant LB medium producer; How plasmid was extracted?

Response: Thank you very much for your advice. I have inserted the manufacturer “(TransGen Biotech, Beijing, China)”. The plasmid was extracted through the plasmid maxi preparation kit(Beyotime, Shanghai, China). Please see M&M 4.1 the yellow part.

Line 243 - medium containing 50 mg of kanamycin - lack of  medium name, producer and volume, lack of kanamycin producer

Response: Thank you very much for your advice. I have inserted the medium name “LB”,producer “(TransGen Biotech, Beijing, China)”and “50 mL” volume, lack of kanamycin producer “(Merck, Shanghai, China)”. Please see M&M 4.2 the yellow part.

Line 251 – How Authors detect signal from secondary antibodies? Please add details, reagents and instruments.

Response: Thank you very much for your advice. I have inserted the information. As follows “The specific operation experiment is as follows: Add SDS loading buffer(Beyotime, Shanghai, China) to the supernatant to be analyzed and boil in boiling water for 10 min to denature the protein. Proteins were then isolated on 12% polyacrylamide gel(Boao Yijie  technology, Beijing, China). The separation gel was stained with Coomassie Bright Blue R-250(Beyotime, Shanghai, China). At the same time, another separation glue was transferred to the PolyScreen PVDF transfer film(Merck, Shanghai, China). The membranes were treated with a closed solution (PBS containing 5% skim milk) for 1 h, incubated with PCV3 porcine positive serum (1:200) (Our laboratory preservation), at room temperature for 2h, washed with PBS 3 times for 5 minutes each time, incubated with goat anti-pig IgG horse radish peroxidase conjugate (1:1000) (Kingsray Biotechnology, Nanjing, China) at room temperature for 1h, washed with PBS 3 times for 5 minutes each time. BeyoECL Plus Western blot Chemiluminescence kit (Beyotime, Shanghai, China) was used for color development after treatment. The Tanon5200 protein imaging analysis instrument(Tanon, Shanghai, China)was then used for signal detection.” Please see M&M 4.2 the yellow part.

Line 253 – lack of ion affinity chromatography instrument/system producer

Response: Thank you very much for your advice. I have added the ion affinity chromatography producer “(Sango, Shanghai, China)”. Please see M&M 4.2 the yellow part.

Line 259 - lack of carbon-coated copper grid producer

Response: Thank you very much for your advice. I have added the carbon-coated copper grid producer “(Zhongjingkeyi, Beijing, China)”. Please see M&M 4.3 the yellow part.

Line 254 - lack of dialysys  bag producer

Response: Thank you very much for your advice. I have added the dialysys  bag producer “(Thermo Fisher, Shanghai, China)”. Please see M&M 4.2 the yellow part.

Line 260 - lack of phosphotungstic acid producer

Response: Thank you very much for your advice. I have added the phosphotungstic acid producer “(Merck, Shanghai, China)”. Please see M&M 4.3 the yellow part.

Line 256 - overnight dialysis was in 4°C?

Response: Thank you very much for your advice. I have added the temperature “4°C”. Please see M&M 4.2 the yellow part.

Line 266 - lack of adjuvant Montanide ISA201 provider

Response: Thank you very much for your advice. I have added the Montanide ISA201 provider“(Seppic, France)”. Please see M&M 4.4 the yellow part.

Line 273 – 281 – lack of PBST, TMB producer, and spectrophotometer name and producer

Response: Thank you very much for your advice. I have added the PBST “(Merck, Shanghai, China)”, TMB producer “(Surmodics, Beijing, China)”, and spectrophotometer name “Biotek ELx800”and producer “(Biotek, Beijing, China)”. Please see M&M 4.5 the yellow part.

Line 284 - lack of  flow tube producer

Response: Thank you very much for your advice. I have added the flow tube producer “(Corning, Shanghai, China)”. Please see M&M 4.6 the yellow part.

Line 287 - 400 g should be calculate to rpm

Response: Thank you very much for your advice. I have changed 400 g to 2000 rpm. Please see M&M 4.6 the yellow part.

Line 243 - 37℃, 220 rpm

Response: Thank you very much for your advice. I have corrected it.

Line 244 – 37℃ for 6-8 hours

Response: Thank you very much for your advice. I have corrected it.

Line 245 - China) was

Response: Thank you very much for your advice. I have corrected it.

Line 293 - China) as follows

Response: Thank you very much for your advice. I have corrected it.

Line 295 - Standard Wells with different concentrations of 50 µL of what?

Response: Thank you very much for your advice. It’s 50 mL the diluted standard from the ELISA kits.

Line 303 - The purified recombinant Cap proteins were coated on?

Response: Thank you very much for your advice. The purified recombinant Cap proteins were coated on ELISA plate with coating solution(1.59 g/L Na2CO3, 2.93 g/L NaHCO3). Please see M&M 4.8 the yellow part.

Line 305, 306, 309 - lack of  bovine serum albumin, serum diluent, TMB, stop solution producers and spectrophotometer name and producer

Response: Thank you very much for your advice. I have added the bovine serum albumin “(Merck, Shanghai, China)”, serum diluent “(Merck, Shanghai, China)”, TMB “(Surmodics, Beijing, China)”, stop solution “(2 M Hâ‚‚SOâ‚„) “  and spectrophotometer name “Biotek ELx800” and producer “(Biotek, Beijing, China).”. Please see M&M 4.8 the yellow part.

Line 318 - lack of  secondary antibody producer

Response: Thank you very much for your advice. I have added the secondary antibody producer “(Kingsray Biotechnology, Nanjing, China)”.

Lin 319 - OD450

Response: Thank you very much for your advice. I have corrected it.

Line 354 - in vitro

Response: Thank you very much for your advice. I have corrected it.